# A New Simplified Autogenous Sinus Lift Technique

**DOI:** 10.3390/bioengineering10050505

**Published:** 2023-04-23

**Authors:** Carlos Aurelio Andreucci, Elza M. M. Fonseca, Renato N. Jorge

**Affiliations:** 1Mechanical Engineering Department, Faculty of Engineering, University of Porto, Rua Dr. Roberto Frias 712, 4200-465 Porto, Portugal; candreucci@hotmail.com; 2LAETA, INEGI, ISEP, Instituto Politécnico do Porto, R. Dr. António Bernardino de Almeida, 4249-015 Porto, Portugal; 3LAETA, INEGI, Mechanical Engineering Department, Faculty of Engineering, University of Porto, Rua Dr. Roberto Frias 712, 4200-465 Porto, Portugal; rnatal@fe.up.pt

**Keywords:** scaffold, sinus lift, dental implant

## Abstract

Oral maxillofacial rehabilitation of the atrophic maxilla with or without pneumatization of the maxillary sinuses routinely presents limited bone availability. This indicates the need for vertical and horizontal bone augmentation. The standard and most used technique is maxillary sinus augmentation using distinct techniques. These techniques may or may not rupture the sinus membrane. Rupture of the sinus membrane increases the risk of acute or chronic contamination of the graft, implant, and maxillary sinus. The surgical procedure for maxillary sinus autograft involves two stages: removal of the autograft and preparation of the bone site for the graft. A third stage is often added to place the osseointegrated implants. This is because it was not possible to do this at the same time as the graft surgery. A new bioactive kinetic screw (BKS) bone implant model is presented that simplifies and effectively performs autogenous grafting, sinus augmentation, and implant fixation in a single step. In the absence of a minimum vertical bone height of 4 mm in the region to be implanted, an additional surgical procedure is performed to harvest bone from the retro-molar trigone region of the mandible to provide additional bone. The feasibility and simplicity of the proposed technique were demonstrated in experimental studies in synthetic maxillary bone and sinus. A digital torque meter was used to measure MIT and MRT during implant insertion and removal. The amount of bone graft was determined by weighing the bone material collected by the new BKS implant. The technique proposed here demonstrated the benefits and limitations of the new BKS implant for maxillary sinus augmentation and installation of dental implants simultaneously.

## 1. Introduction

The maxillary sinuses are air-filled spaces located in both maxilla, lateral to the nasal cavity, superior to the maxillary teeth, inferior to the orbital floor, and anterior to the infratemporal fossa. Furthermore, the maxillary sinus is second to the ethmoid as a source of orbital infection [1]. With an average volume of 12.5 mL, these sinuses are the largest of the paranasal sinuses. The maxillary sinuses are surrounded by a thin bilaminar mucoperiosteal membrane (Schneider’s membrane). This membrane consists of a ciliated pseudostratified columnar epithelium on the luminal side and a single-cell osteogenic periosteal layer on the bony side [2]. In occlusal rehabilitation, the simultaneous placement of osteotome-mediated sinus floor elevation implants with autogenous bone grafting can be considered a successful and predictable treatment approach for deficient posterior maxillary ridges. Different classes of biomaterials, from platelet concentrates to harvested bone and dentin derivatives, are indeed used in sinus lift interventions [3]. This contributes to bone remodeling, resulting in increased endosinus bone gain and a significant reduction in marginal bone loss [4]. Studies systematically reviewed the current evidence on the effect of non-graft versus graft-supported sinus floor elevation on implant survival/failure, endosinus bone gain, crestal bone loss, and bone density around dental implants. Implant survival/failure was the primary outcome. Endosinus bone gain, crestal bone loss, and bone density around dental implants were secondary outcomes. Non-graft maxillary sinus floor elevation appears to be characterised by the new bone formation and high implant survival comparable to bone-graft-supported maxillary sinus floor elevation [5]. There is still no robust evidence on the best techniques for oral rehabilitation with implants in the atrophic maxilla, with or without sinus augmentation. The existence of a wide variety of techniques, implant models, and graft materials in use is evidence of these observations, which are supported by the extensive literature [6,7,8,9,10,11]. All these techniques have their own advantages and disadvantages, ranging from the amount of time involved, to the level of security, cost, and learning curve, such as minimally invasive endoscopic-assisted sinus augmentation [12].

A retrospective study of maxillary sinus lift showed it to be a viable surgical technique allowing implant placement with predictable long-term success regardless of bone graft material type. The success rate of grafts and implants was not affected by the presence of sinus membrane perforation [13]. Another study prospectively evaluated implant status, marginal bone loss, and maxillary sinus floor augmentation outcomes in patients undergoing sinus lift and simultaneous implant placement using bone grafts harvested adjacent to the actual surgical site. Bone grafts can be harvested locally at the site of maxillary sinus augmentation with successful implant placement and healing [14]. One of the most common complications of sinus floor elevation in highly atrophic alveolar ridge augmentation is the perforation of the maxillary sinus membrane. In terms of long-term success, there is no increased risk of implant failure or other persistent complications, such as sinusitis after intraoperative perforation [15,16]. A direct sinus lift, the conventional method of elevating the maxillary sinus, requires surgical access through the lateral wall of the maxilla and elevation of the sinus membrane with the insertion of a bone graft under direct vision. A modified, less invasive method uses a crestal approach without direct membrane visualization, called indirect sinus lift [17].

Tatum in 1970 augmented the posterior maxilla with an autogenous rib and in 1974 developed a Caldwell–Luc surgical technique by fracturing the crest of the alveolus, modifying his own procedure to lift the membrane via a lateral approach [18]. In 1994, Summers developed an internal approach to lift the membrane via an osteotomy, a less invasive procedure for sinus membrane elevation in conjunction with dental implant placement, referred to as osteotome/crestal sinus membrane elevation (OCSME) [7,16,17]. It is indicated for patients with a minimum of 5.0 to 6.0 mm of adequate remaining alveolar bone below the sinus floor. Chen (1996) developed the technique of hydraulic sinus condensation with an osteotomy on the lateral wall of the maxillary ridge [17,19]. The development of hydraulic sinus condensation relies on pressure on the graft material and trapped fluids to hydraulically compress the sinus membrane, creating a blunt force over an expanded area larger than the osteotome tip. In the early days of sinus lift (1974–1979), most grafts were autogenous bone, followed by deproteinised bovine bone mineral (DBBM), and then synthetic grafts, such as beta-tricalcium phosphate (ß-TCP) and calcium phosphosilicate (CPS) [20].

With the introduction and studies of the new bioactive kinetic screw (BKS) dental implant, it is possible to perform a homogeneous autogenous autograft (from the same individual at the same site), transporting the harvested bone to the apical region of the implant [21,22,23,24,25,26]. In the case of implants placed in the atrophic maxilla close to the maxillary sinus, the apical region coincides with the limit defined by the amount of bone and the extension of this sinus. As a natural process of the characteristics and properties of BKS, in which it perforates, screws, collects, and compacts the perforated bone, retaining it in its internal volume, it is possible to lift the sinus membrane using the material’s compression technique in the same simplified surgical procedure. As the transplanted homogeneous autogenous bone is compressed in the internal volume of the BKS, it is transported into the maxillary sinus after the sinus membrane is lifted by pressure and stabilised by the properties of the BKS.

Using a synthetic maxilla and sinus, it is possible to develop and describe an innovative technique for simplified sinus lift using only the concept of simultaneous drilling and screwing, unique inherent characteristics of the BKS, whereby a novel dental implant technique now also simultaneously lifts the maxillary sinus with the pressure provided by the implant apex assembly, densified particulate bone in the internal volume of the BKS, and the controlled insertion torque obtained in the experiment and technique described here. The BKS sinus lift is proven to be a simple and safe technique for immediate indirect sinus lift with homogeneous autogenous bone in cases where a minimum bone height of 4 mm is available. The use of a low rotational speed (60 to 600 rpm) during implantation ensures that implant insertion into the maxillary sinus is safely avoided, and the increased maximum insertion torque achieved by the BKS optimises implant fixation in bone of low density and volume. This technique was developed experimentally for use in further in vivo studies.

## 2. Materials and Methods

In this study, five bioactive kinetic screws (BKS) were machined by Usiform Ltd.a (304 stainless steel) to a length of 10 mm and a diameter of 4 mm with a 15 mm shaft to be connected to the Techdrill Surgical Motor-1 implant motor with a power of 150 watts and a pre-set torque of 45 N/cm for simultaneous drilling and screw insertion. A Lutron TQ 8801 digital torquemeter was used to measure MIT and MRT during implant insertion and removal, and the amount of bone graft was determined by weighing the bone material collected by the new BKS implant using a Brifit precision professional digital mini scale (0.001 to 20.00 g). The technique proposed here, using the innovative BKS dental implant, was performed using a synthetic replica of the constituent bones of the maxillary sinus walls and floor and the sinus membrane, with biomechanical properties such as these structures (Nacional Ossos) used in studies of maxillary sinus augmentation, as seen in Figure 1.

In view of the biomechanical characteristics of the BKS already determined [26], the technique proposed here for maxillary sinus augmentation to place dental implants in the atrophic maxilla consists of the simple simultaneous drilling and screwing of the innovative BKS. After the selection of the dental region to be rehabilitated with a dental implant, the BKS is inserted with a surgical motor using the standard initial drilling procedure for placing dental implants. The difference in the technique lies in the properties and characteristics of the BKS itself, which simultaneously drills, screws, collects, and condenses the bone in its internal volume and, in the case of an atrophic maxilla, transplants the bone into the maxillary sinus without exposing it to the oral environment, as seen in Figure 2.

The MIT torque was measured at the end of implant insertion, as well as its removal and weighing the volume and amount of bone transplanted. After complete insertion of the BKS and augmentation of the maxillary sinus, the sinus membrane was removed to allow direct visualization of the bone graft, as seen in Figure 3.

## 3. Results

The results obtained corroborate with previous studies, adding the values of MIT and MRT measured during the insertion and removal of the BKS, respectively. Considering that one of the properties of BKS is to compact the material in its internal volume, increasing the density of the material applied by 3.45 times [24] densification and solidification of the material was observed, as seen in Figure 4. After perforation of the floor of the maxillary sinus by continuous insertion of BKS, it was noted that the maxillary sinus membrane was lifted without rupture and that the transplanted bone was retained in the BKS implant. Removal of the sinus membrane for direct visualization of the experiment also revealed the presence of bone particles between the sinus floor and this membrane, as seen Figure 3b.

### MIT and MRT

During the insertion of the BKS into the atrophic maxilla, a constant insertion torque was observed, which increased as the entire internal volume was filled with bone. A progressive increase in the rotation was performed automatically by the surgical motor, which was limited to 600 rpm. Even after disruption of the bone floor of the maxillary sinus, the BKS did not reduce the torque and the MIT was measured at 16 N/cm. During the removal of the BKS, the MRT showed a value of 17 N/cm during most of this unscrewing process. The MRT greater than MIT observed in this study was not reported as an occurrence in the literature for bone implants, with it being stated that the removal torque is always less than the insertion torque in both synthetic and natural bone [17,27,28,29].

#### Weight Measurement

The weight of the compacted and densified bone in the internal volume of the BKS was measured directly on a high-precision Brifit scale, thus determining the amount of bone transplanted during the indirect maxillary sinus augmentation procedure proposed here. With the value obtained, it was also possible to determine the bone mass density (*BMD*) of the synthetic bone used in this experiment using Equation (1) determined in a previous study [24]:(1)$$BMD=\frac{Bone\;inside\;BKS}{3.45},$$

Density (*D*) can be calculated using the Equation (2) related to volume (*V*):(2)D=m/V.

The value of bone weight measured inside the BKS was 43 milligrams (mg), as seen in Figure 5. Therefore, the bone mass density found for the synthetic bone used in this study using Equation (1) is 0.44 mg/mm^3^. The bone transplanted into the maxillary sinus was 3.45 times denser (1.53 mg/mm^3^) than the bone from the atrophic maxilla used in this experiment with a BKS inner volume of 28.10 mm^3^. The total volume of bone transplanted without densification is 63.86 mm^3^.

In previous studies, the measurement of the total BKS volume determined was 96.91 mm^3^ [24]. Since the total volume of bone transplanted was 63.86 mm^3^ (bone volume available in the atrophic maxilla), it is possible to determine with a tenth of a millimeter accuracy the minimum bone volume of 33.05 mm^3^ required for complete bone filling in the maxillary sinus to optimise the contact between the bone and the BKS implant to promote the desired sinus augmentation. The bone augmentation achieved was almost twice the minimum required (63.86 mm^3^).

## 4. Discussion

The innovative bioactive kinetic screw (BKS) described here, a simple machine, has the characteristics of a drill, a screw, and a compression machine, combined simultaneously by the application of an insertion torque to fix this device in any material it is capable of drilling, such as bone. It has the characteristics of a self-tapping screw combined with a drill bit with modified flow grooves to hold and compress the material inside. It can be used to drill in the way of a drill bit, to fix in the way of a screw, to collect material for analysis in the way of a biopsy needle, and to graft in the way of a bone graft collector; it can be adapted to any system that uses a screw as a fixation system or a drill bit as an anchor. These combined features allow this new simple machine (BKS) to be used in many ways [21,22,23,25]. As a drill bit, it can be used for more accurate drilling due to the presence of threads that stabilise the drill and help with penetration speed. As there is an established limit to the chip flow of the drilled material, it is also possible to accurately determine the geometry and depth of the hole. The compacted material inside the BKS can be safely removed for analysis, transport, and reuse with minimal handling [26]. It has a higher removal torque than insertion torque and increases the maximum insertion torque. It can be tested for use on bone screws, dental and other limb implants, and prostheses. It can be used as an instrument for measuring the density of materials, including bone [24].

Maxillary sinus augmentation to increase volume and bone quality in the atrophic maxilla became an important tool in oral rehabilitation with dental implants. Autogenous grafts are still considered the gold standard for bone grafting, but they have some disadvantages in their execution. It usually involves another surgical procedure to remove the bone graft from the patient’s donor site and most often a third surgical procedure after an average of 6 months of healing of the grafted bone [4,5]. The proposed new technique, BKS sinus lift, was shown to be similar to conventional dental implant surgery in a single surgical procedure for autogenous bone grafting, sinus lift, and dental implant fixation simultaneously. It also simplified the technique for placing dental implants by using fewer drills to perforate the recipient bone bed site. Care should be taken not to insert the implant into the maxillary sinus, respecting the torque values obtained by simultaneous drilling and screwing of the BKS. This is facilitated by the stability achieved in the simultaneous insertion and drilling of the BKS and its square threads [22].

The increase in MRT in relation to MIT observed in this experiment demonstrates the ability to increase the coefficient of friction between the BKS implant surface and the bone because the densified bone inside it increases the friction between the surfaces by adhesion and cohesiveness between the similar materials bone–implant and bone–bone, respectively [30,31]. In the case of the sinus lift technique proposed here, the hypothesis is that the transferred bone above the floor of the maxillary sinus can also become a mechanical barrier to the removal of the dental implant. The square threads present in the BKS helped to maintain the stability of the implant during and after insertion, including its removal [32]. Regardless of the torque achieved, continuous stability was observed throughout the drilling and removal of the BKS. Although there was no rupture of the sinus membrane in this study, this is a limitation of this experiment because we cannot say that this will happen in clinical practice. Even if the rupture of this membrane were to occur, it is already known in the literature that the rupture of the membrane does not significantly interfere with clinical outcomes [15,16]. We also have the advantage that the technique proposed here is indirect, i.e., without exposing the maxillary sinus to the oral environment and its contaminants.

The densification of the particulate bone (chips) coming from the drill cut was an important observation (Figure 4a,b). It enables the stability of the bone graft used in the maxillary sinus lift. It is expected that the clot that is formed in the natural inflammatory response to the surgical trauma caused by the technique will help to perfuse the graft. This hypothesis should be confirmed in further studies. Since the first theoretical and clinical studies on bone implantation to partially fill with bone the maxillary sinuses, much developed to improve the surgical technique of sinus exposure and augmentation, first with autogenous bone and later with biomaterials [2,3,4,5]. Although the evolution of these techniques contributed to better control the results over the years, the very concept of exposing the maxillary sinus membrane during the surgical procedure increases the risk of infection and other complications [4]. Therefore, authors proposed indirect techniques of maxillary sinus augmentation to avoid sinus exposure, thereby reducing the risk of complications with bone grafting and the success of osseointegration of dental implants [2,19].

The technique of increasing the volume of the maxillary sinus using hydraulic pressure exerted by the bone filling material became an option in cases where there is a vertical height of up to 5 mm in the atrophic maxilla where a dental implant is to be placed [19]. With this concept of indirect augmentation of the maxillary sinus, modified using a new model of dental implant BKS, it is possible to simplify the procedure using autogenous homogeneous bone, i.e., from the patient himself, with bone from the site to be implanted being moved into the maxillary sinus by the biomechanical properties of BKS. In cases where the vertical bone height of the maxillary sinus is less than 4 mm, it is possible to harvest bone from the retromolar region of the patient’s own mandible using the BKS itself as a bone collector [26], as seen in Figure 6, and to transplant the harvested bone in its internal volume to the atrophic maxilla where sinus augmentation for rehabilitation with dental implants is desired. In these cases, an additional surgical procedure is required during the BKS sinus lift surgery, without the need for a new surgery to fix the dental implant.

As the bone quality of the atrophic maxilla is of low density and volume, the proposed study allows simultaneous drilling and screw insertion with the BKS in a single step. In cases where the bone density is higher, an initial guide hole of 1.8 mm diameter can be drilled [24]. This pre-drilling should also be performed (Figure 6b) in cases where transplanted bone from the mandibular retromolar region is used, considering the smaller drilling capacity of the BKS that transports the bone inside. This is desired to increase the bone volume of the atrophic maxilla. The BKS demonstrated the ability to form threaded walls in the perforated bone regardless of the change in rpm (Figure 6b).

Filling the maxillary sinus with synthetic biomaterials, such as hydroxyapatite, was shown to be effective in surgical clinical practice [2,29]. The study proposed here that using synthetic bone became an effective means of demonstrating the feasibility and simplification of the technique of augmenting the maxillary sinus with homogeneous autogenous bone in a single step. Since one of the factors related to the success of osseointegrated implants and maxillary sinus augmentation surgery is related to the specific training of the surgeon, simplifying the technique tends to reduce the risks of performing it [2,33]. It was observed in the literature [2,5,19] that all experiments performed on MIT and MRT measurements and comparisons used mean values for analysis and conclusions. With the technique and experiment described here, the importance of directly comparing MIT and MRT values of the same bone perforation, without comparing other perforations among themselves, became apparent. In this way, it was possible to determine the increase in MRT that could be masked by calculating the average of the values found. Ongoing studies will apply the proposed technique and analyse the benefits of healing and simplification of the sinus floor augmentation technique, comparing the results with other direct and indirect sinus lift procedures.

The main advantage of the BKS is to make the use of fixation screws more efficient by controlling the insertion and stability parameters inherent in its combined characteristics of a modified screw and drill, in which the material from the drill is used to compress in its internal volume, increasing the torque through the internal compression of the applied material, distributing the applied forces in a controlled manner, and increasing the stability of the BKS. The unexpected effect when it was created was the ability to compress material in its internal volume with a precisely estimated amount, to measure the density of the material, to increase the value of the removal torque, and above all, to be able to collect the material when desired, for analysis, transport, and re-insertion or reimplantation in the case of organic materials, such as bone.

## 5. Conclusions

The BKS sinus lift technique proposed and described here was found to be simple and feasible for simplified autogenous transplantation for maxillary sinus augmentation, being similar to the conventional drilling and screwing of dental bone implants. Simultaneous drilling and screwing with square threads provided greater stability for the BKS implant, even with 1/3 of its volume inserted into the maxillary sinus cavity. The experimental tests did not show the “falling into a void” sensation with loss of stability that is usually observed when the implant loses its bone anchorage during implantation. With the limitations described here, the proposed technique proved to be effective in raising the maxillary sinus indirectly, without exposure to the external environment of the sinus membrane. Further studies will evaluate the biological response to this technique.

## Figures and Tables

**Figure 1 bioengineering-10-00505-f001:**
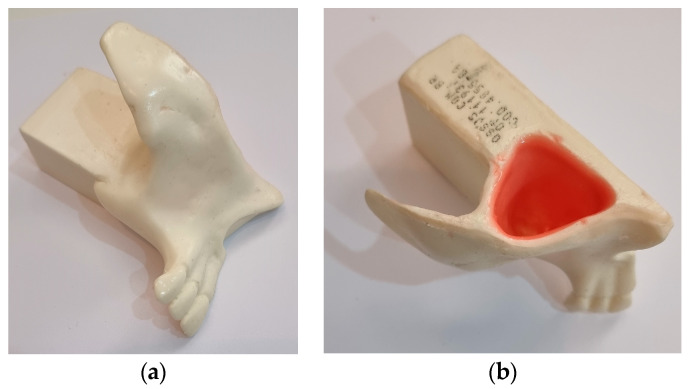
(**a**) Synthetic maxillary sinus bone; (**b**) and maxillary sinus membrane attached to the bone.

**Figure 2 bioengineering-10-00505-f002:**
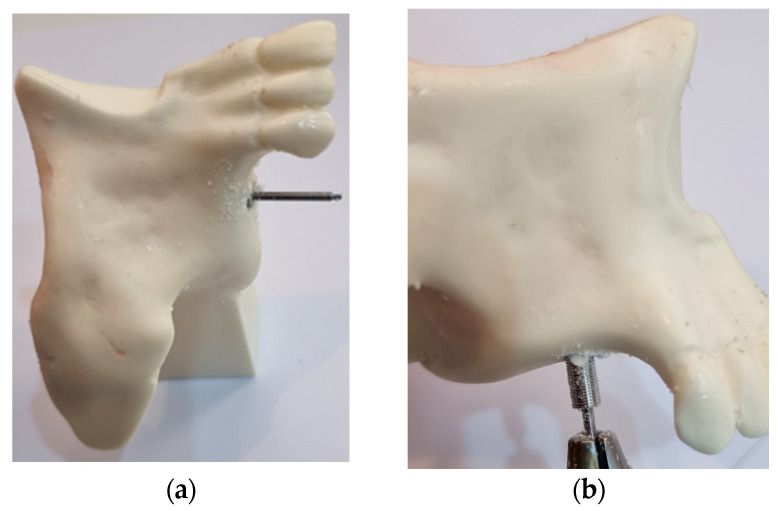
(**a**) BKS totally implanted in atrophic maxilla ready to measure the MRT with the digital torque-meter; (**b**) BKS removed with the compacted collected bone inside to weight measurement.

**Figure 3 bioengineering-10-00505-f003:**
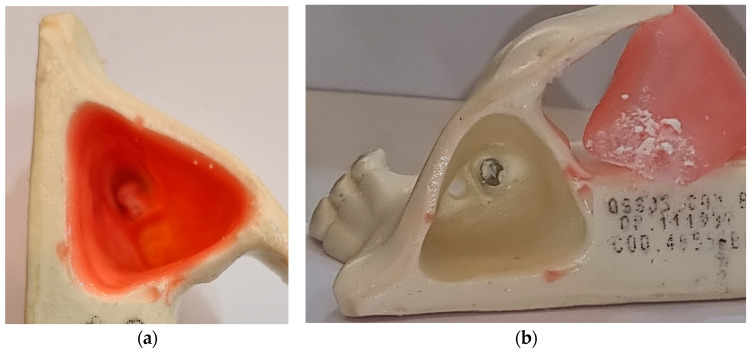
(**a**) Properties of the sinus membrane lifted without rupture by BKS drilling and screwing simultaneously; (**b**) sinus membrane removed showing the bone transplanted between the maxillary sinus floor and the lifted membrane without rupture.

**Figure 4 bioengineering-10-00505-f004:**
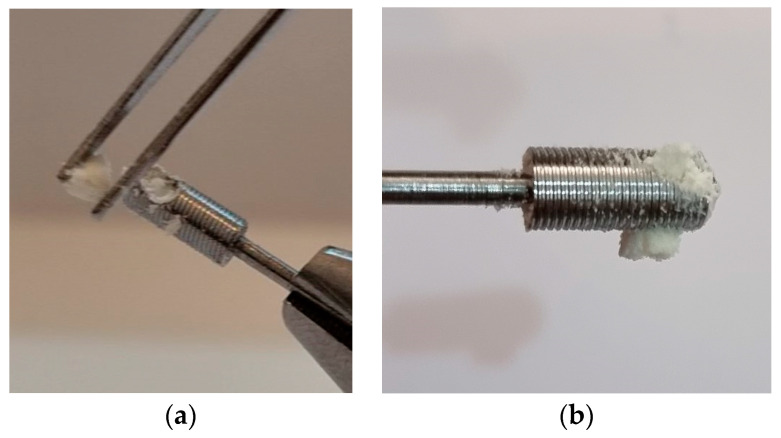
(**a**) and (**b**) Condensation and solidification of the bone particles (chips) inside BKS inherent in its properties of compacting material.

**Figure 5 bioengineering-10-00505-f005:**
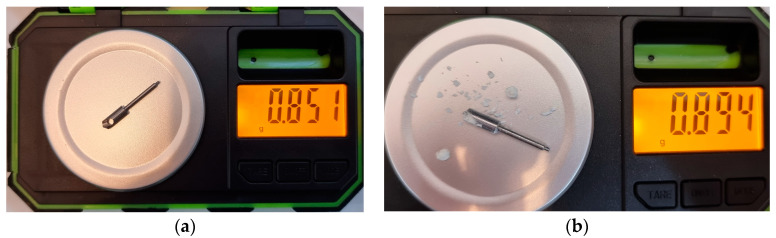
(**a**) BKS original weight; (**b**) BKS and bone weight measurement after total bone removal. The total bone weight is obtained by subtracting the weight of the BKS with bone from the original weight (43 mg).

**Figure 6 bioengineering-10-00505-f006:**
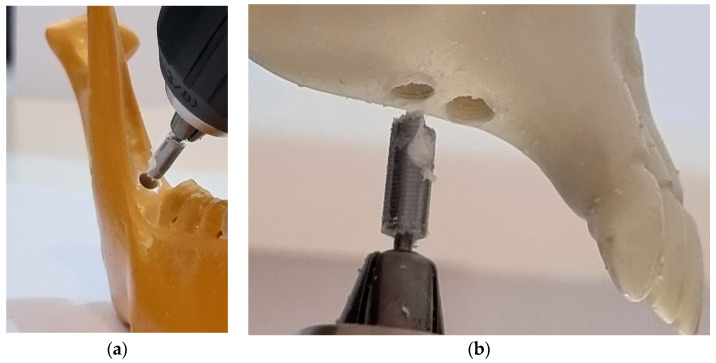
(**a**) Collecting bone from the retromolar region of the synthetic mandible using the BKS itself as a bone collector; (**b**) transplanting the harvested bone in BKS internal volume to the atrophic maxilla.

## Data Availability

No new data were created or analyzed in this study. Data sharing is not applicable to this article.

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
