# Peer review of "A New Simplified Autogenous Sinus Lift Technique"

_bioengineering, 2023, doi:10.3390/bioengineering10050505_

Round 1

Reviewer 1 Report

An interesting first analysis on an innovative method for a maxillary sinus augmentation surgical procedure. A single surgical intervention provides the bone to be grafted -obtained from the maxillary bone itself- while it lifts the sinusal membrane and inserts the active implant surface in the bone wall, thus providing primary stability. An added advantage of this indirect technique is to avoid the risk of exposure of the maxillary sinus to the oral environment by means of a single surgical procedure.

* a minor misshap in line 86 (page 2) has to be corrected: beta-tricalcium phosphate.

Author Response

About the Reviewer’s comments, we are answering their concerns in the following lines. The authors would like again to acknowledge the Reviewer for their work on reading and suggesting improvements to the manuscript. We have addressed all the comments and the changes needed. Changes to the article were identified in green.

We hope that the revisions in the manuscript and our accompanying answers will be sufficient to make our manuscript suitable for publication in Bioengineering.

Comments and Suggestions for Authors

An interesting first analysis on an innovative method for a maxillary sinus augmentation surgical procedure. A single surgical intervention provides the bone to be grafted -obtained from the maxillary bone itself- while it lifts the sinusal membrane and inserts the active implant surface in the bone wall, thus providing primary stability. An added advantage of this indirect technique is to avoid the risk of exposure of the maxillary sinus to the oral environment by means of a single surgical procedure.

  1. * a minor misshap in line 86 (page 2) has to be corrected: beta-tricalcium phosphate

ANSWER – The authors thank the reviewer. The corrections were introduced.

Reviewer 2 Report

Comments on Andreucci et al:

The aim of this manuscript is to analyze a new autogenous sinus lift techniquesimple and feasible for simplified autogenous transplantation for maxillary sinus augmentation.

This manuscript shows rich content, providing a deep insight for some works: the study is within the journal’s scope, and I found it to be well-written, providing sufficient information. Even if the manuscript provides an organic overview, with a densely organized structure and based on well-synthetized evidence, there are some suggestions necessary to make the article complete and fully readable. For these reasons, the manuscript requires major changes.

Please find below an enumerated list of comments on my review of the manuscript:

INTRODUCTION:

LINE 36: Furthermore, the maxillary sinus is second to the ethmoid as a source of orbital infection (see, for reference: RaponiI, Giovannetti F, Buracchi M, Priore P, Battisti A, Scagnet M, Genitori L, Valentini V. Management of orbital and brain complications of sinusitis: A practical algorithm. Craniomaxillofac Surg. 2021 Dec;49(12):1124-1129. doi: 10.1016/j.jcms.2021.09.005). 

LINE 38: The authors should discuss about maxillary sinus floor augmentationa surgical procedure that allows the rehabilitation of atrophic edentulous posterior maxilla also crucial to dental implant procedure (see, for reference: Giovannetti F, Raponi I, Priore P, Macciocchi A, Barbera G, Valentini V. Minimally-Invasive Endoscopic-Assisted Sinus Augmentation. CraniofacSurg. 2019 Jun;30(4):e359-e362. doi: 10.1097/SCS.0000000000005365).

LINE 43: The authors should highlight that different class of biomaterials, from platelet concentrates to harvested bone and dentin derivates are indeed used in sinus lift interventions (see, for reference: Bernardi, S.; Macchiarelli, G.; Bianchi, S. Autologous Materials in Regenerative Dentistry: Harvested Bone, Platelet Concentrates and Dentin Derivates. Molecules 202025, 5330. https://doi.org/10.3390/molecules25225330).

The main topic is interesting, and certainly of great clinical impact. Overall, the contents are rich, and the authors also give their deep insight for some works.

As regards the section of methods, there is a specific and detailed explanation for the methods used in this study: this is particularly significant, since the manuscript relies on a multitude of methodological and statistical analysis, to derive its conclusions. The methodology applied is overall correct, the results are reliable and adequately discussed.

The conclusion of this manuscript is perfectly in line with the main purpose of the paper: the authors have designed and conducted the study properly. Conclusions are well written and present an adequate balance between the description of previous findings and the results presented by the authors.

In conclusion, this manuscript is densely presented and well organized, based on well-synthetized evidence. The authors were lucid in their style of writing, making it easy to read and understand the message, portrayed in the manuscript. Besides, the methodology design was appropriately implemented within the study. However, many of the topics are very concisely covered. This manuscript provided a comprehensive analysis of current knowledge in this field. Moreover, this research has futuristic importance and could be potential for future research. However, major concerns of this manuscript are with the introductive section: for these reasons, I have major comments for this section, for improvement before acceptance for publication. The article is accurate and provides relevant information on the topic and I have some major points to make, that may help to improve the quality of the current manuscript and maximize its scientific impact. I would accept this manuscript if the comments are addressed properly.

Author Response

About the Reviewer’s comments, we are answering their concerns in the following lines. The authors would like again to acknowledge the Reviewer for their work on reading and suggesting improvements to the manuscript. We have addressed all the comments and the changes needed. Changes to the article were identified in green.

We hope that the revisions in the manuscript and our accompanying answers will be sufficient to make our manuscript suitable for publication in Bioengineering.

Comments on Andreucci et al:

The aim of this manuscript is to analyze a new autogenous sinus lift technique, simple and feasible for simplified autogenous transplantation for maxillary sinus augmentation.

This manuscript shows rich content, providing a deep insight for some works: the study is within the journal’s scope, and I found it to be well-written, providing sufficient information. Even if the manuscript provides an organic overview, with a densely organized structure and based on well-synthetized evidence, there are some suggestions necessary to make the article complete and fully readable. For these reasons, the manuscript requires major changes.

Please find below an enumerated list of comments on my review of the manuscript:

INTRODUCTION:

  1. LINE 36: Furthermore, the maxillary sinus is second to the ethmoid as a source of orbital infection (see, for reference: RaponiI, Giovannetti F, Buracchi M, Priore P, Battisti A, Scagnet M, Genitori L, Valentini V. Management of orbital and brain complications of sinusitis: A practical algorithm. J Craniomaxillofac Surg. 2021 Dec;49(12):1124-1129. doi: 10.1016/j.jcms.2021.09.005).

ANSWER – The authors thank the reviewer. The reference was introduced.

  1. LINE 38: The authors should discuss about maxillary sinus floor augmentation, a surgical procedure that allows the rehabilitation of atrophic edentulous posterior maxilla also crucial to dental implant procedure (see, for reference: Giovannetti F, Raponi I, Priore P, Macciocchi A, Barbera G, Valentini V. Minimally-Invasive Endoscopic-Assisted Sinus Augmentation. J CraniofacSurg. 2019 Jun;30(4):e359-e362. doi: 10.1097/SCS.0000000000005365).

ANSWER – The reference was introduced.

  1. LINE 43: The authors should highlight that different class of biomaterials, from platelet concentrates to harvested bone and dentin derivates are indeed used in sinus lift interventions (see, for reference: Bernardi, S.; Macchiarelli, G.; Bianchi, S. Autologous Materials in Regenerative Dentistry: Harvested Bone, Platelet Concentrates and Dentin Derivates. Molecules 2020, 25, 5330. https://doi.org/10.3390/molecules25225330).

ANSWER – The extension revision to improve readability was introduced.

The main topic is interesting, and certainly of great clinical impact. Overall, the contents are rich, and the authors also give their deep insight for some works.

As regards the section of methods, there is a specific and detailed explanation for the methods used in this study: this is particularly significant, since the manuscript relies on a multitude of methodological and statistical analysis, to derive its conclusions. The methodology applied is overall correct, the results are reliable and adequately discussed.

The conclusion of this manuscript is perfectly in line with the main purpose of the paper: the authors have designed and conducted the study properly. Conclusions are well written and present an adequate balance between the description of previous findings and the results presented by the authors.

In conclusion, this manuscript is densely presented and well organized, based on well-synthetized evidence. The authors were lucid in their style of writing, making it easy to read and understand the message, portrayed in the manuscript. Besides, the methodology design was appropriately implemented within the study. However, many of the topics are very concisely covered. This manuscript provided a comprehensive analysis of current knowledge in this field. Moreover, this research has futuristic importance and could be potential for future research. However, major concerns of this manuscript are with the introductive section: for these reasons, I have major comments for this section, for improvement before acceptance for publication. The article is accurate and provides relevant information on the topic and I have some major points to make, that may help to improve the quality of the current manuscript and maximize its scientific impact. I would accept this manuscript if the comments are addressed properly.

ANSWER – All comments were introduced to improve the quality of the manuscript.

Reviewer 3 Report

The title promises every oral surgeon: oh, this is a new simple and safe surgical method that I can use tomorrow. But it is only a proposal for a possible new surgical method that has been developed on a model (!)! Since apparently no animal experiments or clinical trials are available or mentioned so far that would convince a clinician, for the sake of fairness you should change the title to "Proposal of a new....".

Otherwise, there is nothing to criticize in the perfect description of bone-grafting and hypothetical (!) healing and reinforcement of the atrophied sinus wall under the theoretically (!) intact mucosa.

But you should write the abbreviation BKS (Bioactive Kinetic Screw) in the abstract (line 19) and then under Materials (line 88 or 114) and also state the manufacturer. 

Congratulations for this ingenious technique and best wishes for successful clinical trials !

Author Response

About the Reviewer’s comments, we are answering their concerns in the following lines. The authors would like again to acknowledge the Reviewer for their work on reading and suggesting improvements to the manuscript. We have addressed all the comments and the changes needed. Changes to the article were identified in green.

We hope that the revisions in the manuscript and our accompanying answers will be sufficient to make our manuscript suitable for publication in Bioengineering.

Comments and Suggestions for Authors

The title promises every oral surgeon: oh, this is a new simple and safe surgical method that I can use tomorrow. But it is only a proposal for a possible new surgical method that has been developed on a model (!)! Since apparently no animal experiments or clinical trials are available or mentioned so far that would convince a clinician, for the sake of fairness you should change the title to "Proposal of a new....".

Otherwise, there is nothing to criticize in the perfect description of bone-grafting and hypothetical (!) healing and reinforcement of the atrophied sinus wall under the theoretically (!) intact mucosa.

But you should write the abbreviation BKS (Bioactive Kinetic Screw) in the abstract (line 19) and then under Materials (line 88 or 114) and also state the manufacturer.

Congratulations for this ingenious technique and best wishes for successful clinical trials !.

ANSWER – The authors thank the reviewer. All changes were introduced to improve the quality and the title was changed. Thanks for your words.

Round 2

Reviewer 2 Report

Manuscript can be now accepted